# Factors Influencing Farmers' Vertical Collaboration in the Agri-Chain Guided by Leading Enterprises: A Study of the Table Grape Industry in China

**Wen Li [1], Chenying Liu [1,\*], Qizhi Yang [2], Yulan You [1], Zhihang Zhuo [1] and Xiaolin Zuo [1]**

[1] School of Management, Jiangsu University, Zhenjiang 212013, China
[2] School of Agricultural Equipment Engineering, Jiangsu University, Zhenjiang 212013, China
\* Correspondence: liuchenying@163.com

**Abstract:** Leading agricultural enterprises attracting farmers to participate in vertical collaboration within the industry chain can expedite the process of agricultural industrialization and help achieve rural revitalization. This study focuses on table grape growers in the Yangtze River Delta region of China as the research subjects. Instead of examining the impact of specific elements individually on farmers' involvement in vertical collaboration within the industry chain, this study emphasizes the combination of multiple factors influencing farmers' engagement. Employing a fuzzy-set qualitative comparative analysis and adopting a configurational perspective, this research investigates how six factors—growers' age, ratio of sales income to household income, production scale, market distance, financial support, and technical support—combine to influence farmers' participation in vertical collaboration within the industry chain. This study reveals that leading enterprises can indeed promote farmers' participation in vertical cooperation within the agricultural chain, and four pathways are identified. Based on these findings, three configurations are summarized: risk-averse, resource-constrained, and burden-alleviating. Specific strategies and recommendations for targeting each of these configurations are proposed based on the findings, along with policy suggestions for regulating the six factors, both by the enterprises themselves and by the government.

**Keywords:** agricultural leading enterprise; farmers' participation; agricultural industrialization; qualitative comparative analysis; table grape



## 1. Introduction

In the current context, prioritizing the development of agriculture and rural areas, promoting the modernization of agriculture, accelerating the collaboration of agricultural industry chains, and building a complete agricultural industry chain are the top priorities of the rural revitalization strategy [1]. Vertical collaboration refers to economic cooperation between upstream and downstream enterprises in the supply chain and farmers in the supply chain. It encompasses all vertically interdependent and mutually cooperative production and sales activities among stakeholders in the supply chain, including market transactions, various forms of contracts, and complete integration [2]. At present, vertical collaboration can be categorized into three basic types: market transaction relationships, integration relationships, and contractual relationships [3]. Vertical collaboration between enterprises and farmers has a profound impact on food quality and safety [4,5]. Back in the early 21st century, Hobbs and Young (2000) put forward the framework and mechanisms for vertical collaboration in agricultural value chains [6]. Agricultural value chains are intricate and encompass value chains, organizational chains, information chains, and supply chains [7–11]. Various actors participate in these chains, including enterprises, governments, industry organizations, dispersed farmers, and large-scale producers, each with distinct characteristics [12]. Tarabukina (2021) emphasized the importance of agricultural clustering and vertical integration for ensuring competitiveness and the sustainable

development of enterprises [13]. Sun and van der Ven (2020) highlighted the close collaboration between farmers and enterprises as an effective approach to achieving sustainable aquaculture practices in Asia [14]. Trifković (2014) explored the interaction between food standards and vertical collaboration in the Vietnamese basa fish industry [15]. Kaminski et al. (2020) attempted to integrate small-scale farmers with other stakeholders in the value chain through inclusive business models (IBMs) to foster deep collaboration [16]. Ba et al. (2019) focused on the vertical coordination between exporters and farmers in the Mekong Delta's rice value chain through contract farming to upgrade the industry [17]. Lezoche et al. (2020) argued that effective organization and connectivity among actors in the agricultural supply chain, facilitating real-time information exchange, can enhance overall chain performance [11].

The modern construction of agricultural industry chain systems primarily focuses on exploring the vertical extension and integration of supply chains and value chains, involving multiple stages such as agricultural production, distribution, processing, and sales, which can be categorized into three phases: pre-production, production, and post-production [18]. Carvajal et al. (2018) propose that enhancing the integration capacity of supply chains can help improve the competitiveness of enterprises [19]. Garnett (2020) argues that establishing close vertical collaboration relationships in the food industry chain contributes significantly to the recovery from the impacts of COVID-19 and plays a crucial role in building supply chain resilience [20]. Research by Ramos et al. (2021) suggests that the integration of internal and external vertical collaboration in the specialty coffee supply chain directly influences supply chain resilience [21]. Wang et al. (2023) point out that vertical integration in the industry chain is beneficial for pig enterprises to achieve complementary interests and sustainable development [22]. They identify previous asset specificity, the legal system environment, market demand, and transaction frequency as the main influencing factors. Bhat et al. (2022) proposed an Agri-SCM-BIoT (agriculture supply chain management using blockchain and the Internet of things) architecture in an attempt to use blockchain and IoT technologies to improve the traceability issues in the agro-industrial chain [23]. Li and Huang (2020) conducted an analysis of the impact of the Internet on agricultural value chains and proposed that blockchain technology is advantageous for the integrated development of agricultural value chains and enhances the traceability of agricultural products [24].

Since the reform and opening up in China, and against a background of industrial integration, the national table grape industry has experienced rapid development. In all grape-producing countries, grapes are utilized for various purposes, primarily fresh consumption, wine production, and raisin manufacturing. Globally, table grapes are predominantly cultivated in China, India, Turkey, Uzbekistan, and the European Union. Notably, China stands as the largest producer of table grapes worldwide, consistently accounting for approximately 50% of the global production, establishing itself as the foremost producer and consumer of table grapes on a global scale [25]. The Xiahei grape contraception incident attracted much attention, demonstrating that as the consumption market of fresh fruits continues to heat up, consumers are paying more and more attention to grape quality and planting methods [26–28]. As the largest table grape production base in the Yangtze River Delta region of China, the Grape Town of Jurong also suffered significant losses during the Xiahei incident. To implement the "Guiding Opinions of the State Council on Promoting the Revitalization of Rural Industries" and the spirit of the Central No. 1 Document for 2023, table grape agricultural leading enterprises are accelerating their upstream expansion to the place of origin, actively trying to cooperate with planting households and forming an agricultural industry chain integration model of "company + farmers" [29,30]. However, at the current stage, the organizational relationship of this model is loose, the degree of organization of planting households is low, and the cooperation is not close, which facilitates opportunistic behavior by both parties [31]. In order to spread and encourage the leading enterprises more effectively, improve the cooperative relationship of the division of labor and cooperation with farmers, and build an

intensive production base that gathers individual planting households around the core of a leading enterprise, the leading enterprise needs to enhance its value-added services and the coordination and integration capabilities it provides to the industry chain [32–36]. As important subjects in the upstream part of the industry chain, farmers' vertical cooperation behavior and choice willingness directly affect the level of vertical cooperation in the agricultural industry chain [37–40]. Therefore, it is necessary for leading enterprises to study the factors influencing individual planting households' willingness to participate in the vertical cooperation of the industry chain.

Previous studies have analyzed the factors influencing farmers' participation in vertical collaboration within agricultural value chains and their willingness to participate. Wang et al. (2021) conducted a case study on the kiwifruit industry in Shaanxi, China, and found that farmers' participation in agricultural value chains has a significant impact on poverty reduction, encouraging active cooperation with enterprises to promote technological and facility development [41]. German et al. (2020) argue that the structural factors of inclusive enterprises limit the space for inclusiveness, hindering farmers' participation in vertical collaboration within value chains [42]. Bizikova (2020) suggests that the services provided by farmers' organizations (FOs) facilitate a deep engagement of farmers in collaborative activities within value chains [43]. Martins et al. (2019) found positive effects of horizontal relationships in the Brazilian pig farming supply chain on vertical collaboration between farmers and buyers, achieved through an improved information exchange between farmers and buyers [44]. Von Loeper et al. (2016) employed a system dynamics model to understand the influence of banks on farmers' participation in value chain collaboration [45], thus drawing attention to the widespread interest in inclusive finance [46]. Adegbite and Machethe (2020) argue that digital finance and gender-inclusive agricultural finance innovations will greatly contribute to the willingness of Nigerian farmers to collaborate within agricultural value chains [47]. Abdul-Rahaman (2020) conducted research on rice farmers in the northern part of Ghana and found that age, access to credit opportunities, labor supply, the ability to sell to corporate buyers, and association membership significantly influenced farmers' participation in vertical collaboration within value chains [48]. Mossie et al. (2020) examined apple and mango farmers and found that age, market distance, education level, and contact with buyers had significant impacts on farmers' participation in vertical collaboration within value chains [49]. Gurmis and Melese (2022) explored the factors influencing avocado farmers' involvement in market decision making and market participation in the Kaffa region of Ethiopia [50]. These studies indicate that farmers' willingness is influenced by various types of uncertainty.

It can be observed that many scholars have paid considerable attention to vertical collaboration in the agricultural industry chain, particularly focusing on agricultural industrialization, the level of agricultural organization, and the role of agricultural leading enterprises. However, there is a lack of attention paid to the guiding role of agricultural leading enterprises in vertical collaboration within the industry chain. Therefore, this paper takes vertical collaboration in the industry chain as the focal point, emphasizing the leading role of agricultural enterprises in engaging farmers in vertical collaboration within the industry chain, thereby strengthening the connection between upstream farmers and enterprises in the industry chain. Research has addressed the influencing factors of farmers' participation in vertical collaboration within the industry chain for the agricultural and forestry sectors. However, the analytical perspective has primarily revolved around the farmers' viewpoint, predominantly utilizing quantitative descriptive statistical analysis methods. With advancements in research methods, it has become evident that it is not only individual factors which impact the results; the interactions of different factors can collectively influence the outcomes.

Consequently, based on the research data collected from Jurong Grape Town in the Yangtze River Delta region, this article constructs and analyzes the participation mode of individual planting households in vertical cooperation within the industry chain from the perspective of leading agricultural enterprises, and explores the multiple interactive

relationships among the various factors affecting the participation of individual planting households in tight vertical cooperation within the industry chain from the perspective of configurational theory.

## 2. Materials and Methods

### 2.1. Mechanisms Analysis

This investigation revealed that the scale and level of the grape industry in the Yangtze River Delta region have a leading position nationwide. Jurong Grape Town, as the largest table grape production base in the Yangtze River Delta region, occupies an area of 11.18 square kilometers, with a grape cultivation area of over 10,000 mu and involving 1097 households engaged in grape cultivation [51]. It is a nationally renowned grape industry village. The town hosts three international and provincial-level leisure agriculture and rural tourism demonstration sites and has established cooperatives with distinctive features. It stands out as the planting base with the most grape varieties and the highest level of standardization for fresh grape production in the country. For this article, integrating the three stages of table grape production—pre-production, production, and post-production—allowed a thorough exploration of the structure of production, processing, storage, transportation, and sales, enabling construction of an operational model of the existing table grape industry chain, as illustrated in Figure 1.

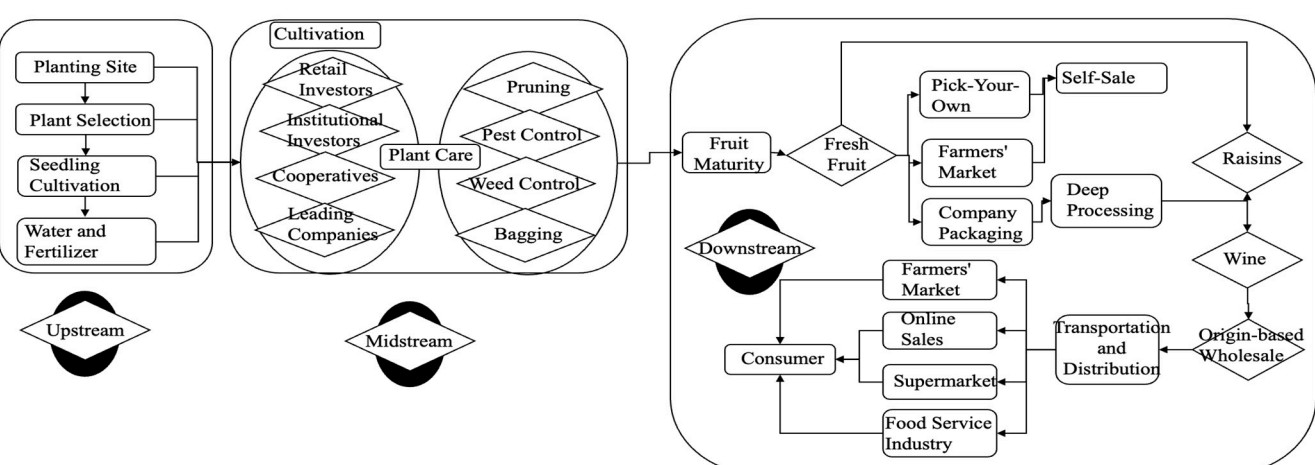

**Figure 1.** The operating model of the table grape industry chain.

This diagram illustrates that, from the perspective of grape growers, there exists vertical collaboration in the pre-production, production, and post-production processes of grapes, with interconnections between the upstream and downstream entities in the industry chain. Prior to grape cultivation, growers need to select a planting base. In the case of grape growers in Jurong, the land is mostly owned by the growers themselves. Some large-scale growers choose to lease additional land from other scattered landowners, while others entrust their land to leading enterprises through cooperatives. Fertilizers and seedlings can be obtained through self-production or purchase. Some growers in collaboration with leading enterprises use specified agricultural materials provided by the companies. During the grape cultivation process, leading enterprises offer technical assistance, but many individual growers choose to manage their own respective planting areas. Some larger growers may employ a small number of laborers to assist in management. Additionally, most of the pesticides and tools for pest control and weed removal during grape cultivation are purchased by growers themselves from the market. After the grapes ripen, growers are responsible for the harvesting, packaging, transportation, and sale of fresh grapes. A few growers engage in mutual assistance and cooperation. However, due to self-interest and a lack of responsibility among some growers, only a few scattered growers have established mutually beneficial relationships with cooperatives and leading

enterprises, ultimately resulting in inconsistent grape quality. It is evident that the operational model of the grape industry in Jurong exhibits characteristics of diversity, complexity, and fragmentation. Numerous such issues persist within the vertical collaboration of the industry chain.

### 2.2. Identification of Influencing Factors

The agricultural industry chain is a network structure that serves various aspects of farming and agricultural product sales, with significant correlations and closeness. Based on a literature review and field research, the formation of tightly integrated vertical cooperation within the industrial chain is not a straightforward process, as it presents challenges stemming from several factors [52]. These include differences in technology use and environmental resources during the pre-production stage, the individual and household characteristics of farmers, production-related features during the production stage, and intricate market dynamics and the broader socio-economic backdrop with government policy support during the post-production stage. The lack of close vertical cooperation within the industrial chain results from the combined and mutually influencing effects of these multiple factors. Six factors were selected as antecedents influencing Jurong table grape farmers' participation in vertical co-operation in the industry chain: market distance, ratio of sales income to household income, age, production scale, technical support, and financial support.

Market distance has a significant impact on the vertical collaboration of growers in the industry chain [53]. The distance from the nearest grape trading market is directly related to the transportation cost and sales risk for grape growers. If the grower is closer to the grape sales market, the transportation cost will be lower and the grape sales profit will be higher. They will be more willing to transport the grapes to the nearby market and sell them at a relatively higher profit based on price comparisons. However, if the distance from the grape trading market is greater, the cost of transporting grapes to the market will be higher, and the transportation cost that is passed on to the grape sales price will also be correspondingly higher. This will lower the price competitiveness and introduce a greater risk of unsold grapes in the market. As close vertical collaboration between the production and sales links can solve the grape sales problem more effectively, growers tend to participate in close vertical collaboration when the distance to the grape trading market is far. The higher the proportion of sales income to the total household income, the higher the growers' expected income and the higher the risk they face. In this case, if the enterprise or cooperative can provide help in the grape production and sales processes and help them to avoid some risks, the growers' willingness to participate in close vertical collaboration will be more robust [54]. The older the growers, the less likely they are to accept and understand new sales methods and policies, and the lower their willingness will be to participate in vertical collaboration in the industry chain [54]. The larger the production scale of growers, the higher their willingness will be to participate in vertical cooperation in the industry chain [52]. The larger the scale of grape cultivation, the greater the risk faced by growers. In order to stabilize the grape sales channel and stabilize sales income, growers will have a higher enthusiasm for participating in close vertical collaboration. The participation of growers in close vertical collaboration within the industry chain is influenced by factors such as resources, including funding and technology [54]. The ease of obtaining funds and technology has an impact on the willingness of growers to choose the vertical collaboration mode [52]. Grape cultivation requires a certain investment of funds and access to technology. If the enterprise or cooperative can provide support for funding and technology for farmers, their willingness to participate in close vertical collaboration will be greater.

This article introduces configuration theory into the study of the factors affecting grape growers' participation in vertical collaboration in the industry chain, and takes Jurong Grape Town as the research case to explore how these six factors of market distance, sales income as a proportion of household income, age, production scale, technical support,

and financial support combine to affect the vertical collaboration of Jurong table grape growers in the industry chain. It also studies how these factors combine to affect vertical collaboration and it analyzes their impact paths. This leads to policy recommendations for local leading enterprises to further guide growers' participation in vertical collaboration in the industry chain.

### 2.3. Fuzzy-Set Qualitative Comparative Analysis Method

Fuzzy-set qualitative comparative analysis (fsQCA) is a research method that combines qualitative and quantitative approaches. It is based on the theory and methodology of qualitative comparative analysis (QCA), integrating Boolean algebra and set theory into a binary variable QCA technique [55]. It offers a new research perspective for addressing complex causal relationships [56], combining the advantages of both quantitative and qualitative analysis. By comparing a certain number of cases, it analyzes and summarizes the concurrent causal relationships among different sets. Specifically, it explores how different combinations of variables may influence multiple cases to produce a similar phenomenon, aiming to understand the process of complex societal issues arising from multiple concurrent causal inductions from a holistic perspective [57]. Compared to crisp-set qualitative comparative methods and multi-value-set qualitative comparative analysis, fsQCA can better prevent information loss during data transformation, enhance data precision, and more accurately identify effects caused by changes in antecedent conditions [58]. Due to its ability to transform causal relationships into complex causality characterized by fuzziness, asymmetry, and equifinality, fsQCA can address issues related to the partial membership of sets and more precisely capture the influence of changing conditional variables on outcome variables [59].

This study employed the fsQCA method, primarily considering the existing research that indicates that to thoroughly investigate the influencing factors of farmers' participation in vertical collaboration in the industry chain, it is inadequate to solely analyze the independent effects of individual variables [55]. It is imperative to approach the research holistically, exploring the combined effects of various complex variables. Additionally, fsQCA transforms fuzzy sets into truth tables, retaining the advantages of analyzing qualitative data, dealing with limited diversity, and simplifying configurations using truth table analysis. This endows the research with the dual attributes of qualitative and quantitative analysis [60]. Additionally, this study employed the fsQCA 3.0 software to set qualitative anchors, allowing for a precise calibration of variable membership degree and outcome membership degree for cases in the set through programmatic calculations. Necessary analyses were conducted for individual condition variables, constructing truth tables to investigate the impact of combinations of condition variables on the result variable. Finally, an analysis of the configuration of conditions was performed to assess sufficiency based on the effects of configurations of different condition variables on the outcome variable [58].

### 2.4. Data Source

The data for this study were collected through a household survey conducted in Jurong Grape Town, Jiangsu Province, in August 2022. Three villages were selected for the survey, Dingzhuang Village, Xisong Village, and Nanshanzi Village, and farmers engaged in grape cultivation were surveyed in each village. Jurong Grape Town, as one of the three demonstration sites for leisure agriculture and rural tourism in the Yangtze River Delta, serves as the grape planting base with the greatest variety and highest level of standardization nationally. In order to ensure the representativeness of the samples, the criteria for selecting sample growers in this study were as follows: First, the sample growers were engaged in grape production activities and identified as grape growers. Second, the sample growers had a relatively high rate of commercialization of grape production, reaching 70%. Third, the sample growers needed to be farming in the grape production base in Jurong. In each survey location, grape growers were randomly selected based on the criteria mentioned above for household surveys. The survey included basic information

about the household head and other individuals in the household, family circumstances, grape production and sales, and the grape grower's participation in vertical collaboration in the production and sales processes. The surveyors were graduate students with extensive experience in rural grassroots surveys, ensuring the quality of the survey to a certain extent. A total of 150 questionnaires were distributed and collected for the survey, resulting in 141 valid questionnaires. The respondents in this study were generally aged 50 and above, had low levels of education, and included 13 households of small-scale farmers each having a planting area of over 20 mu, as well as a total of 128 individual scattered growers. The distribution of the sample is shown in Table 1.

**Table 1.** Sample distribution.

| Planting Area | Villages | Number of Distributed Questionnaires | Number of Questionnaires Collected | Response Rate of the Survey Questionnaire |
|---|---|---|---|---|
| Jurong City | Dingzhuang Village | 45 | 42 | 93.33% |
| | Xisongzhuang | 67 | 63 | 94.03% |
| | Nanshanzi | 38 | 36 | 94.74% |
| Total | | 150 | 141 | 94.00% |

## 3. Results

Before conducting the empirical analysis, it was necessary to perform a test of the reliability and validity of the data. The specific results of the test were as follows.

### 3.1. Reliability and Validity Test

SPSS 20.0 software was used to analyze the reliability, convergent validity, and discriminant validity of the scales used in this study. The reliability measurement results reflected the stability of the data, while those for validity indicated the degree of closeness between the measurement results and the intended goals. The values of the Cronbach's $\alpha$ coefficient and CR for the variables were both greater than 0.7, and the AVE values for all variables were greater than 0.5, indicating that the reliability and validity of the variables were relatively good. The reliability and validity tests are shown in Table 2.

**Table 2.** Reliability and validity tests.

| Variable | Cronbach's $\alpha$ | CR | AVE | A | SI/HI | PS | MD | FS | TS |
|---|---|---|---|---|---|---|---|---|---|
| A | 0.845 | 0.910 | 0.753 | 0.884 | | | | | |
| SI/HI | 0.856 | 0.919 | 0.772 | 0.638 | 0.849 | | | | |
| PS | 0.853 | 0.915 | 0.768 | 0.675 | 0.617 | 0.892 | | | |
| MD | 0.889 | 0.981 | 0.892 | 0.554 | 0.496 | 0.564 | 0.963 | | |
| FS | 0.796 | 0.927 | 0.776 | 0.607 | 0.527 | 0.769 | 0.796 | 0.832 | 0.924 |
| TS | 0.825 | 0.903 | 0.829 | 0.596 | 0.623 | 0.632 | 0.628 | 0.681 | 0.796 |

Note: A—age; SI/HI—ratio of sales income to household income; PS—production scale; MD—market distance; FS—financial support; TS—technical support; CR—composite reliability; AVE—average variance extracted.

### 3.2. Data Calibration

Calibration is the process of transforming variables into a set form and assigning set membership degrees to cases [59]. In this study, the fsQCA3.0 software was used to set qualitative anchors, and a precise calibration of variable membership degrees and outcome membership degrees in the set was performed through program calculations, with 1 indicating complete membership of the variable and 0 indicating complete non-membership of the variable.

Since Likert's five-point scale was used to collect the data in this study and the membership values in fsQCA range from 0 to 1, before conducting the qualitative comparative analysis, the raw data needed to be converted into values on a 0–1 scale. After considering the distribution of the scores for each variable, the calibration values for complete

membership, crossover, and complete non-membership were determined. Specifically, the corresponding anchor points for each variable were "age" of 5, 4, 2.75; "Ratio of Sales Income to Household Income" of 4, 1.5, 1; "production scale" of 3.25, 2, 1; "market distance" of 4.5, 2.5, 1; "financial support" of 5, 3.5, 1; "technical support" of 5, 4, 2; and "willingness to participate in vertical cooperation" of 1, 0.5, 0. The calibrated membership degrees for each variable are shown in Table 3.

**Table 3.** Fuzzy set membership (partial).

| Sample | A | SI/HI | PS | MD | FS | TS | N |
|--------|------|-------|------|------|------|------|------|
| 1 | 0.80 | 0.24 | 0.18 | 0.25 | 0.51 | 0.12 | 0.95 |
| 2 | 0.76 | 0.36 | 0.32 | 0.11 | 0.50 | 0.25 | 0.95 |
| 3 | 0.95 | 0.85 | 0.05 | 0.32 | 0.95 | 0.75 | 0.05 |
| 4 | 0.35 | 0.11 | 0.50 | 0.43 | 0.95 | 0.79 | 0.95 |
| 5 | 0.96 | 0.85 | 0.45 | 0.54 | 0.50 | 0.55 | 0.05 |
| 6 | 0.5 | 0.50 | 0.50 | 0.50 | 0.12 | 0.93 | 0.95 |
| 7 | 0.13 | 0.44 | 0.44 | 0.55 | 0.12 | 0.79 | 0.95 |
| 8 | 0.76 | 0.80 | 0.80 | 0.81 | 0.55 | 0.67 | 0.95 |
| 9 | 0.73 | 0.42 | 0.42 | 0.19 | 0.22 | 0.85 | 0.95 |
| 10 | 0.42 | 0.38 | 0.16 | 0.21 | 0.32 | 0.98 | 0.05 |
| 11 | 0.28 | 0.97 | 0.97 | 0.97 | 0.47 | 0.93 | 0.95 |
| 12 | 0.69 | 0.91 | 0.91 | 0.92 | 0.55 | 0.56 | 0.95 |
| 13 | 0.17 | 0.94 | 0.94 | 0.94 | 0.85 | 0.73 | 0.95 |
| 14 | 0.36 | 0.93 | 0.93 | 0.55 | 0.56 | 0.55 | 0.05 |
| 15 | 0.80 | 0.27 | 0.40 | 0.32 | 0.55 | 0.40 | 0.95 |

*3.3. Single-Condition Necessity Analysis*

This study assessed the necessity of each individual condition variable's relationship with the result variable through a necessary condition analysis. When the consistency in the analysis result is 1, it indicates that the condition variable is a completely necessary factor for the result, and there exists a perfect subset relationship between the condition variable and the result variable. Before conducting the fuzzy-set truth table program analysis, the consistency threshold for the necessary conditions was set to 0.9. If more than 90% of the result variable belongs to a certain condition variable, then the necessary condition for the result variable is this condition variable. In the equation, $X_i$ represents the membership degree of the conditional variable (or conditional combination) and $Y_i$ represents the membership degree of the corresponding result variable [58].

$$\text{consistency}(X_i \leq Y_i) = \sum (\min(X_i, Y_i)) / \sum X_i \tag{1}$$

$$\text{consistency}(X_i \leq Y_i) = \sum (\min(X_i, Y_i)) / \sum Y_i \tag{2}$$

The analysis was conducted using fsQCA 3.0 software, and the results are presented in Table 4. The sufficiency and necessity of the individual conditions affecting grape growers' willingness to participate in vertical collaboration in the industry chain were examined. The consistency threshold for individual conditions was consistently below 0.9, indicating that individual conditions alone cannot constitute sufficient conditions for the result variable. Additionally, the coverage coefficient for individual conditions was also below 0.9, indicating that individual conditions alone cannot constitute necessary conditions for the result variable. Therefore, this study further analyzed the combined paths of conditional variables and their influence on grape growers' willingness to participate in vertical collaboration.

**Table 4.** Results of necessary condition analysis.

| Antecedent Condition Variables | Consistency | Coverage |
|:---:|:---:|:---:|
| A | 0.842119 | 0.835742 |
| ~A | 0.494459 | 0.784844 |
| SI/HI | 0.766922 | 0.590725 |
| ~SI/HI | 0.571441 | 0.839150 |
| PS | 0.517476 | 0.695544 |
| ~PS | 0.530887 | 0.729037 |
| MD | 0.483376 | 0.752298 |
| ~MD | 0.554987 | 0.871486 |
| FS | 0.421995 | 0.815118 |
| ~FS | 0.616368 | 0.763464 |
| TS | 0.569480 | 0.794293 |
| ~TS | 0.468031 | 0.843308 |

Note: "~"—negation/absence of condition.

### 3.4. Configuration Analysis

After conducting the necessary analysis on individual condition variables, this study proceeded to analyze the sufficiency of configurations formed by different combinations of condition variables for the result variable. The intermediate solution was utilized to determine the number of configurations and the conditions they encompass. This was then combined with the parsimonious solution to distinguish between the core condition and contributory conditions [58]. Solid circles represent the presence of a condition variable, while hollow circles indicate its absence. A large circle represents a core condition variable, a small circle represents a contributory condition variable, and the absence of a circle denotes that the condition variable may or may not be present. The fsQCA 4.0 software was used for standardized analysis, and results of the configuration analysis of the combinations forming the influencing factors of grape growers' participation in vertical collaboration in the industry chain is presented in Table 5.

**Table 5.** Combined pathways of conditional variables.

| Condition Variable | Combination of Factors Influencing the Participation of Growers in Vertical Collaboration in the Agricultural Industry Chain | | | |
|:---:|:---:|:---:|:---:|:---:|
| | N1 | N2 | N3 | N4 |
| A. | ● | ⊗ | | ● |
| SI/HI | ● | ● | ● | ⊗ |
| PS | ⊗ | ● | ● | ⊗ |
| MD | ⊗ | | ● | ● |
| FS | ● | ● | ● | |
| TS | ● | ● | | ● |
| Raw Coverage | 0.245382 | 0.237349 | 0.318875 | 0.433333 |
| Unique Coverage | 0.0682731 | 0.0738956 | 0.0417671 | 0.0618474 |
| Consistency | 0.988621 | 0.9485964 | 0.976879 | 0.935287 |
| Solution Coverage | 0.747586 | | | |
| Solution Consistency | 0.964153 | | | |
| Algorithm | Quine–McCluskey | | | |

Note: ●/•—core/contributory condition present; ⊗/⊗—core/contributory condition absent.

Based on Table 5, the overall consistency of the configuration was 0.964, indicating that the six factors explain the variation in farmers' choices to participate in vertical cooperation to a high degree. The total coverage was 0.747, meaning that the research findings can cover 74.7% of the cases. The consistency of all configurations was higher than the accepted standard of 0.8, indicating that these configurations have a good subset relationship with farmers' willingness to participate in vertical cooperation. This demonstrated that the antecedent conditions have good explanatory power for the dependent variable (strong

willingness to participate in vertical cooperation). Based on the analysis, this study identified four pathways that significantly influence farmers' willingness to participate in vertical cooperation in the grape industry.

Pathway N1 (A*M*~S*~L*V*E) indicates that if farmers are older, have a higher percentage of grape sales income in the household income, have a smaller cultivation scale, are closer to the market, and face greater difficulties in obtaining financial and technical support, they will be more willing to participate in vertical cooperation. This pathway had the highest consistency index and the strongest explanatory power.

Pathway N2 (~A*M*S*V*E) indicates that if farmers are younger, have a higher percentage of grape sales income in the household income, have a larger cultivation scale, and face greater difficulties in obtaining financial and technical support, they will be more willing to participate in vertical cooperation.

Pathway N3 (M*S*L*V) indicates that if farmers have a higher percentage of grape sales income in the household income, have a larger cultivation scale, and are farther away from the market, they will be more willing to participate in vertical cooperation.

Pathway N4 (A*~M*~S*L*E) indicates that if farmers are older, have a lower percentage of grape sales income in the household income, have a smaller cultivation scale, are farther away from the market, and face greater difficulties in obtaining technical support, they will be more willing to participate in vertical cooperation. The explanatory power of the pathways was ranked as N1 > N3 > N2 > N4.

*3.5. Result Discussion*

Based on these four pathways, three configurations that affect table grape growers' participation in vertical cooperation in the industry chain can be summarized as follows.

Configuration 1: Risk avoidance driven. Pathway N1 refers to grape growers upstream in the industry chain who are relatively old and rely mainly on grape sales for family income. Although these growers are involved in small-scale grape cultivation, they face significant risks in planting, maintenance, harvesting, and sales because these activities are mainly carried out individually. Similarly, pathway N3 indicates that some growers, although they may have a large cultivation scale, are too busy to attend to other businesses and rely heavily on grape sales for family income. Because they are far from the market, the transportation cost and the difficulty of obtaining information about sales increase significantly, which increases the risk of cultivation. Therefore, growers are more willing to participate in vertical cooperation in the industry chain to avoid risks.

It is necessary for leading agricultural enterprises to ensure that farmers feel at ease. This requires enterprises to provide more benefits and guarantees for elderly farmers, better profit-sharing mechanisms, market development support, technical training, and financial support. At the same time, government agencies can regulate the obligations and rights of both agricultural leading enterprises and farmers in vertical cooperation to prevent the occurrence of "opportunistic" behavior and promote the formation of a long-term win–win mechanism and a stable and close cooperation relationship between both sides. The government also needs to establish a risk-sharing mechanism, jointly bear agricultural production risks with enterprises and farmers, reduce their risk pressure, and improve confidence and security.

Configuration 2: Resource scarcity driven. Pathway N2 indicates that some young growers, such as large-scale growers or family farms, rely mainly on table grape sales for income, and lack sufficient funds and technical support to enable grape deep processing. The limited sales channels and the greater difficulty in brand development make it challenging to increase income. Therefore, they have a strong desire to participate in vertical cooperation in the industry chain to safeguard their income.

Resource-scarcity-driven pathways can be used to guide growers to enhance their willingness to participate in vertical collaboration in the industry chain and also to identify growers who are highly motivated to participate in such collaboration. Agricultural leading enterprises can cooperate effectively with the government, make reasonable use of relevant

policies for regulation, and ensure that agricultural policies are effectively implemented. Agricultural leading enterprises can also leverage the government's credibility, policies, and funding advantages to influence the specialization, scaling, and standardization of table grape cultivation. In addition, relevant government departments can urge leading enterprises to provide timely technical consultation and training, and demonstration services to growers. The government can also optimize the allocation and utilization of land and water resources, improve agricultural production efficiency, and create a better development environment for agricultural leading enterprises and growers.

Configuration 3: Burden reduction driven. Pathway N4 indicates that the older the grower, the weaker their ability to cultivate grapes, and the smaller their cultivation scale. In addition to income from grape sales, they have other income sources, such as vegetable cultivation and sales, animal husbandry income, and pension income. At the same time, because they are far from the market, their transportation and sales abilities are weak. They have a strong desire to participate in vertical cooperation in the industry chain to reduce their burden.

The sharing of production is worth emphasizing. For growers with limited abilities in planting, transportation, and sales, agricultural leading enterprises can provide services such as sharing agricultural materials and agent planting. Meanwhile, relevant government departments can strengthen policy guidance, introduce a series of policies to encourage cooperation between agricultural leading enterprises and growers, continue to promote the "cooperative society-led enterprise + growers" model, and explore new models to promote deep cooperation in the industry chain. Through the reasonable division of labor and resource integration, growers' burdens can be reduced.

Through induction and analysis of the four pathways, it can be seen that grape growers have a strong willingness to participate in vertical cooperation in the industry chain under the combined influence of six factors: age, the proportion of grape sales to family income, cultivation scale, distance from the market, difficulty in obtaining funding support, and difficulty in obtaining technical support. This not only strengthens the head enterprise's supervision and support for its own base and individual cultivation but also provides clues for enterprises about how to target growers. Moreover, it can be seen that the willingness of growers to participate in vertical cooperation in the industry chain is influenced by multiple factors, which effectively explains the complexity of the configurations and improves the credibility of analyzing problems based on the configuration perspective.

## 4. Conclusions

This study took the table grape cultivation industry in Jurong, a region within the Yangtze River Delta, as a case study. It employed the fuzzy-set qualitative comparative analysis (fsQCA) method to analyze and explore the configurational pathways of growers' participation in close vertical collaboration within the industry chain. While providing constructive suggestions, to some extent, for guiding growers' participation in vertical collaboration within the industry chain, there are certain limitations in this regard.

In this research, the configurational fsQCA primarily focused on the combined effects of six condition variables (age, production scale, market distance, the proportion of sales income to household income, and technical and financial support) concerning growers' participation in vertical collaboration within the industry chain. For the diversity and optimality of the condition variables, further research and exploration are needed to include other condition variables in the configurational analyses.

Furthermore, this study identified four pathways and three configurations. To enhance the accuracy and reliability of the research results and provide more effective guidance for practical applications, future research could explore the modeling and simulation of optimal pathways based on system dynamics. This approach would allow the observation of the evolution process and outcomes of the model, analyzing the feasibility of the optimal pathway for leading enterprises to guide growers' participation in vertical collaboration

within the industry chain. Consequently, practical and feasible recommendations could be presented to the leading enterprises based on this analysis.

**Author Contributions:** Conceptualization, C.L. and X.Z.; methodology, C.L., Z.Z., Y.Y. and X.Z.; software, C.L. and X.Z.; validation, C.L. and Y.Y.; writing—original draft preparation, X.Z.; writing—review and editing, C.L. and W.L.; supervision, W.L. and Q.Y.; project administration, W.L.; funding acquisition, W.L. and Q.Y. All authors have read and agreed to the published version of the manuscript.

**Funding:** This research was funded by the Independent Innovation Funds for Agricultural Science and Technology (Major) Project of Jiangsu Province, grant number CX(21)1007; the Soft Science Project of Zhenjiang City, grant number RK2022007; and the Key Project of Agricultural Equipment Discipline of Jiangsu University, grant number NZXB20210101.

**Institutional Review Board Statement:** Not applicable.

**Data Availability Statement:** The data that are presented in this study are available from the corresponding author upon request. The data are not publicly available due to privacy restrictions.

**Conflicts of Interest:** The authors declare no conflict of interest.

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
