# Peer review of "Factors Influencing Farmers’ Vertical Collaboration in the Agri-Chain Guided by Leading Enterprises: A Study of the Table Grape Industry in China"

_agriculture, doi:10.3390/agriculture13101915_

Round 1

Author Response

1.Summary

Thank you very much for taking the time to review this manuscript. Second, we are pleased to have the opportunity to revise and resubmit this improved manuscript.

In our response to the questions posed, we provided a comprehensive overview of the changes made in response to the review suggestions and constructive comments. In this context, we have revised the paper based on your extensive and insightful comments. We believe that this revised version of the paper is much more substantial.

We responded to your comments point by point. After explaining it to you, we uploaded the complete manuscript with the changes as an attachment and highlighted the changes in yellow.

We hope that we have met your expectations, but of course we stand ready to make any further adjustments or give you any further consideration.

2. Point-by-point response to Comments and Suggestions

Comments 1: The article is interesting and the subject of the paper fits the field of interest of AGRICULTURE. However, it has several typos and grammatical errors that require a careful English language revision.

Response 1: Thank you for pointing this out. We agree with this comment. Therefore, to address the typos and grammatical errors in the article, we have carried out a detailed revision using the language editing service provided by DMPI. The revisions are marked in red in the submitted annex.

Comments 2: The quality of research – in terms of methodological rigor, analytical skills and relevance - it is medium / high. The method adopted by the author also presents elements of originality useful for the scientific debate. There is not always clarity in the exposition: some passages are confused and some considerations are not adequately supported, while some repetitions are present, probable synthesis of a more complex research work of clear scientific value.

Response 2: Agree. We have made some major adjustments to the order of presentation in the introduction to make the logic clearer. Firstly, it points out the importance of vertical collaboration in the industrial chain for rural development, then it introduces vertical collaboration and the industrial chain as well as the current research status of the two, then it introduces the grape industry and draws out the role of leading enterprises in the vertical collaboration of the industrial chain, and finally it introduces the previous research on the participation of farmers in the vertical collaboration of the industrial chain.

Comments 3: The abstract contains the main sections of the paper and describes findings, ma It is suggested that you review the abstract as the objective of the research does not emerge clearly.

Response 3: Couldn't agree more with your point of view. In the abstract we added the research objective: we would like to explore how the combination of multiple factors influences the participation of farmers in vertical collaboration in the chain. Lines 12-14 in the submitted annex are highlighted in yellow.

Comments 4: The introduction describes the main aim of the study. In the Introduction the Author states the contribution of the manuscript. In the introduction, the innovative aspect of the paper does not clearly emerge: it is suggested to emphasize the innovative contribution of the paper with respect to the existing literature.

Response 4: Thank you for pointing this out. We added a paragraph to the introduction highlighting the paper's innovative contribution to the existing literature. Yellow highlighting was done on lines 12-14 in the submitted attachment. Yellow highlighting was done on lines 137-151 in the submitted attachment.

Comments 5: The review of the literature is exhaustive but must be more contextualized with respect to the topic of the article. Furthermore, it is suggested to insert some references in support of some concluding remarks.

Response 5: Thanks to your suggestion. Based on your suggestions, we have reorganised the literature review. In addition, we have added the definition, origin, and components of vertical collaboration to make it more relevant to the topic(line32-40). Finally, we have added some new references to support the concluding remarks (line98 and line 103).

Comments 6: The theoretical references appropriately referred to in the work and denote mastery of the subject. The model used needs, to improve the fluidity of the exposure, of further details with particular reference to the definition the model. Tables and figures are appropriate.

Response 6: Agree. We provide a clearer explanation before the model is built (lines 169-173); in explaining the model, we go into more detail about vertical collaboration between dragon-head enterprises and farmers and the problems (lines 179-194).

Comments 7: The results, described with appropriate comments on tables and figures, are affected by a certain hasty in the final part of the paragraph, with evident repetitions and overlaps with the conclusions: improve this section.

Response 7: Thank you very much for raising this point. We have removed the overlap between results and conclusions, and added the limitations of the present research, and possible further research directions.

Reviewer 2 Report

The article raises a very interesting and current research topic. Despite the fact that the article uses statistical methods appropriate to the adopted subject matter, the article contains several very important shortcomings. The most important of these shortcomings is the lack of a clearly formulated publication purpose (including specific objectives) and the main hypothesis and related sub-hypotheses. Such a situation causes the authors to analyze who knows what? The reader can only guess what the authors verify and what goal they want to achieve. The above remarks also lead to a fundamental remark related to the abstract. A well-written abstract should include the goal and hypotheses. Unfortunately, I don't see it in this abstract, so I suggest a correction.

In addition to the above critical remarks, I believe that the article covers interesting and current topics, is written in a manner with an acceptable editorial level, and that methods adequate to the analyzed issues were used, and conclusions were drawn based on the analyzes carried out, but as already mentioned, the conclusions must necessarily relate to to the goal and hypotheses adopted in the publication.

Author Response

1.Summary

Thank you very much for taking the time to review this manuscript. Second, we are pleased to have the opportunity to revise and resubmit this improved manuscript.

In our response to the questions posed, we provided a comprehensive overview of the changes made in response to the review suggestions and constructive comments. In this context, we have revised the paper based on your extensive and insightful comments. We believe that this revised version of the paper is much more substantial.

We responded to your comments point by point. After explaining it to you, we uploaded the complete manuscript with the changes as an attachment and highlighted the changes in yellow.

We hope that we have met your expectations, but of course we stand ready to make any further adjustments or give you any further consideration.

2. Point-by-point response to Comments and Suggestions

Comments 1: The article raises a very interesting and current research topic. Despite the fact that the article uses statistical methods appropriate to the adopted subject matter, the article contains several very important shortcomings. The most important of these shortcomings is the lack of a clearly formulated publication purpose (including specific objectives) and the main hypothesis and related sub- hypotheses. Such a situation causes the authors to analyze who knows what? The reader can only guess what the authors verify and what goal they want to achieve. The above remarks also lead to a fundamental remark related to the abstract. A well-written abstract should include the goal and hypotheses. Unfortunately, I don't see it in this abstract, so I suggest a correction.

Response 1: Firstly, thank you for your pertinent suggestions in this revision. we have added in the abstract the research objective: we want to investigate how the combination of multiple factors influences the participation of farmers in the vertical collaboration of the industrial chain. This was highlighted in yellow on lines 12-14 in the submitted annex. Secondly, we added a paragraph in the introduction to explain the gaps in the previous research and to account for the purpose of this study. It has been highlighted in yellow in lines 137-151 in the submitted annex. Furthermore, we have made some major adjustments to the order of presentation in the introduction to make the logic clearer. It points out the importance of vertical collaboration in the industrial chain for rural development, then it introduces vertical collaboration and the industrial chain as well as the current research status of the two, then it introduces the grape industry and draws out the role of leading enterprises in the vertical collaboration of the industrial chain, and finally it introduces the previous research on the participation of farmers in the vertical collaboration of the industrial chain.

Regarding the construction of the main hypothesis and related sub-hypotheses. We believe that hypotheses are not necessary for the fuzzy set qualitative comparative analysis approach. We constructed a model based on the interviews and distilled the influencing factors, followed by a QCA analysis to determine which combinations of factors influenced the results. The purpose of a configurational analysis is to reveal how multiple factors are configured to influence outcomes, so it is difficult to assume which combinations of factors will have an impact on the results before analysing them.

Reviewer 3 Report

This article is interesting, but need some improvements as follows:

1.       Methods should be written  briefly and clearly in the abstract.

2.       What is the definition of Vertical Collaboration here?

3.       What is table grape production? Reader outside China might be not familiar with this term. Please explain it.

4.       The vertical collaboration is for plantation crop or including food crop and horticulture?

5.       What is the research gap of this study?

6.       Why did the study only conducted in Jurong city?

7.       What the meaning of on the Table 4?

8.       I don’t see there are any empirical evidences on the discussion.

 Could you please present the characteristics of your respondents?

Author Response

1. Summary

Thank you very much for taking the time to review this manuscript. Second, we are pleased to have the opportunity to revise and resubmit this improved manuscript.

In our response to the questions posed, we provided a comprehensive overview of the changes made in response to the review suggestions and constructive comments. In this context, we have revised the paper based on your extensive and insightful comments. We believe that this revised version of the paper is much more substantial.

We responded to your comments point by point. After explaining it to you, we uploaded the complete manuscript with the changes as an attachment and highlighted the changes in yellow.

We hope that we have met your expectations, but of course we stand ready to make any further adjustments or give you any further consideration.

2. Point-by-point response to Comments and Suggestions

Comments 1: Methods should be written briefly and clearly in the abstract.

Response 1: Thanks for the suggestion, in lines 14-18 of the submitted attachment, we have re-edited the description of the methodology in the summary.

Comments 2: What is the definition of Vertical Collaboration here?

Response 2: Thanks to your comments, the definition of vertical collaboration is given in introduction (lines 32-39 highlighted in yellow) and in the model analysis (lines 176-197 highlighted in yellow). It refers to the market, fiduciary, and contractual behaviours carried out by leading enterprises and growers before, during, and after production.

Comments 3: What is table grape production? Reader outside China might be not familiar with this term. Please explain it.

Response 3: Thank you for your valuable comments. A description of the table grape production is given in the Mechanisms Analysis section (lines 89-91 and 161-169).

Comments 4: The vertical collaboration is for plantation crop or including food crop and horticulture?

Response 4: Thank you for your question. The vertical collaboration is including food crop and horticulture. Because the vertical collaboration in this paper is chain-wide vertical collaboration (lines 32-39 and 176-197), food crop and horticulture are both in the pre-production activities.

Comments 5: What is the research gap of this study?

Response 5: We added a paragraph (lines 137-151) to the introduction to illustrate a gap in previous research: previous research has not explored how the combination of multiple factors influences farmers' participation in vertical chain collaboration.

Comments 6: Why did the study only conducted in Jurong city?

Response 6: Because the scale and level of the grape industry in the Yangtze River Delta region has ranked among the top in the country, Jurong Grape Town is the largest production base of table grapes in the Yangtze River Delta region, and it is a nationally renowned village of grape industry. It has three international and provincial leisure agriculture and rural tourism demonstration sites, established a distinctive cooperative union, and is representative of the planting base with the most varieties of grapes and the highest level of standardisation of table grapes in the country. The importance and status of Jurong Grape Town is expressed in lines 89-91, 161-169 and 288-295 of the article.

Comments 7: What the meaning of ~ on the Table 4?

Response 7: “~”means Negation/Absence of condition. A note has been added to line 363 of the text, thank you for the suggestion.

Comments 8: I don’t see there are any empirical evidences on the discussion.

Response 8: The three configurations are summarized through four pathways. This analysis aids in understanding the motivations and reasons for farmers' participation in industrial chain cooperation, as well as the various challenges they face. Different configurations may require the leading enterprises to adjust the six factors to varying degrees to facilitate the vertical cooperation of farmers in the industrial chain, in order to achieve common interests and goals.

Comments 9: Could you please present the characteristics of your respondents?

Response 9: In order to ensure the representativeness of the samples, the criteria for selecting sample growers in this study were as follows: First, the sample growers were engaged in grape production activities and identified as grape growers. Second, the sample growers had a relatively high rate of commercialization of grape production, reaching 70%. Third, the sample growers needed to cover the grape production base in Jurong (lines 291-295). The respondents in this study were generally aged 50 and above, had low levels of education, and included 13 households of small-scale farmers with a planting area of over 20 mu, as well as a total of 128 individual scattered growers (lines 302-305).

Reviewer 4 Report

General comments:

The paper presents an important topic, the vertical integration of the grape supply chain in China. The methodology (fsQCA) is a good choice, applied correctly. The presentation of the research results is sound, and the overall quality of the paper is good.

However, there are a few issues that should be improved before the paper is published. One main weakness - though it should be easy to correct, is the specific QCA terminology, which may not be well known for the average reader, therefore the authors should provide a more detailed explanation of the methodology and the specific terms used (see details below).

The language of the paper is good (only a few typos are found), the references are correct, and reasonably wide. Therefore I suggest minor revisions.

Details:

Abstract.

At the end of the Abstract the authors should briefly indicate the novelty - or the research gap - that the paper adds to the current state-of-the art.

Introduction: 

The introduction is good, it summarises the importance of the topic, the previous empirical results in agricultural supply chain and vertical integration, including international and Chinese examples. There is a clear description of the need for vertical integration in the table grape production chain in China.  Some minor points are the following:

Line 37-38: a reference should be added to ""Guiding Opinions of the State Council on Promoting the Revitalization of Rural Industries" and the spirit of the Central No.1 Document for 2023"

At the end of the Introduction (after line 128) the authors should add a short paragraph in which they explicitly give the research gap that the paper is addressing, or the novelty of the results.

Section 2 (Identification of Mechanisms...)

Line 156-157: Here the authors present Figure 1- but it is too small, and difficult to read. Enlarge the two panel, place them under each other, so that a larger size is allowed. Also, give the source of the figure (own construction, or adapted from some literature?)

Line 162: "...field research, The formation of  ...", the capital T should be lowcase t.

Section 3: Data and methods.

Here the authors introduce the QCA- fsQCA methodology. This section is a bit too short. This methodology has a specific procedure, terminology, and software. Therefore, a more detailed presentation would be needed here. 

A small technical issue: line 224 refers to existing studies, but without references. Please add some references here. 

Before Table 1 (line 252): when presenting the sample of 150 farmers, please explain how this low number of respondents can be considered a representative sample (or representative sample of what area, group, etc.) - and therefore to what extent can the results be considered representative for the grape growers of the analysed region.

In Table 2 (and later in Tab 3-4-5) instead of the letters A-M-S...  etc. please choose some short variable names more suggestive of the content or meaning of the actual variables. I suggest, that you give a list of your variables, with the value ranges and descriptive statistics before the analysis (somewhere before Table 2). Then you can refer to this list (or new table) in the legends of the other tables presenting your results.

In the footnote of Table 2 please give the full name of CR and AVE (composite reliability and Average variance ...)

4.2 Data calibration

Line 269: give a precise reference for the fsQCA software. Or even better, place the reference of the software in the Methodology section, where you should also give the brief summary of how this software operates, what inputs and outputs are used, what specific terms are used (e.g. condition variable, subset, result variable, truth table, consistency threshold, membership degree...) A very good description of the terminology can be found in the following paper:

A VOCABULARY FOR QCA by Roel Rutten and Claude Rubinson, published on compasss.org December 2022 (it is easy to find it online).

Another good reference for the fsQCA methodology is in: Avoiding Common Errors in QCA: A Short Guide for New Practitioners, by Claude Rubinson & Lasse Gerrits & Roel Rutten & Thomas Greckhamer (published on July 22, 2019) - again an easy web-search can find this paper for you.

Another very good paper that can help you give a brief, but reasonably detailed description of the fsQCA methodology is: Presenting qualitative comparative analysis: Notation, tabular layout, and visualization - by Claude Rubinson, in Methodological Innovations, May-August 2019: 1–22, at https://doi.org/10.1177/2059799119862110

These are easily downloadable, open-access sources, and could help you to summarise the methodology with more details, and sufficient detail.

I also suggest, that you present some of your results not only in tabular form, but as diagrams, too, for ideas see again the mentioned paper: Presenting qualitative comparative analysis: Notation, tabular layout, and visualization - by Claude Rubinson.

4.3 Results 

The section title (in line 316) should be 4.4. Results

Line 318-319: please explain the meaning of "parsimonious solution, intermediate solution, and complex solution", in non-technical, easily understandable form here.

Table 5, line 331-334: please explain what do you mean by "core condition, and peripheral condition" as mentioned in the footnotes of the table. Also, the markers of large black dot and small black dot should be changed to something more clearly distinguishable, and the use of blank space for the absence of peripheral condition should also be changed to something visible.

Line 336: you refer to six configurations, but you have only four ones. 

Line 337: you say that "the total coverage is 0.847, ... " but it is actually 0.747 - please correct.

4.4 Results Discussion  

Line 364: the title should be changed to 4.5 Discussion

Results are well explained, the final 3 configurations are logically constructed - however, I suggest that the authors should use a different word, not configuration (of which they have already presented four) - maybe three pathways (?), or some other word.

In this section the authors should also present some comparisons of their findings to other, previous empirical results of QCA-based research, or results referring to agricultural vertical integration influencing factors...

Another minor point here:

Line 337: At the beginning of the line the lowcase "it" should be started with a capital "I", It...

5. Conclusions

Conclusions are sound, but at the end of this section the authors should mention the limitations of the present research, and possible further research directions.

With these improvements, I consider the paper needing minor revisions.

Language is good and clearly understandable, only a few typos are in the paper (for details see above).

Author Response

1.Summary

Thank you very much for taking the time to review this manuscript, your suggestions are invaluable to us. Secondly, we are pleased to have the opportunity to revise and resubmit this improved manuscript.

We have revised your suggestions one by one in the article and we believe that the revised paper is more informative. We have uploaded the revised full manuscript as an attachment with notes corresponding to the suggestions you made.

We hope we have met your expectations, but of course, we are always ready to make further adjustments or give you further consideration.

Round 2

Reviewer 1 Report

The revision are made and the manuscript is sufficient for publication

Reviewer 3 Report

Thank you for revising the article largely. I can accept it.